# Sirtuin 1, Visfatin and IL-27 Serum Levels of Type 1 Diabetic Females in Relation to Cardiovascular Parameters and Autoimmune Thyroid Disease

**DOI:** 10.3390/biom11081110

**Published:** 2021-07-28

**Authors:** Magdalena Łukawska-Tatarczuk, Edward Franek, Leszek Czupryniak, Ilona Joniec-Maciejak, Agnieszka Pawlak, Ewa Wojnar, Jakub Zieliński, Dagmara Mirowska-Guzel, Beata Mrozikiewicz-Rakowska

**Affiliations:** 1Clinical Department of Internal Diseases, Endocrinology and Diabetology, Central Clinical Hospital of the Ministry of Interior and Administration, Wołoska 137, 02-507 Warsaw, Poland; magdalukawska89@gmail.com (M.Ł.-T.); edward.franek@cskmswia.pl (E.F.); 2Department of Diabetology and Internal Diseases, Medical University of Warsaw, Banacha 1a, 02-097 Warsaw, Poland; leszek.czupryniak@wum.edu.pl; 3Mossakowski Medical Research Centre, Polish Academy of Sciences, Adolfa Pawińskiego 5, 02-106 Warsaw, Poland; a.pawlak1@wp.pl; 4Department of Experimental and Clinical Pharmacology, Centre for Preclinical Research and Technology (CePT), Medical University of Warsaw, Banacha 1b, 02-097 Warsaw, Poland; Ilona.joniec@wum.edu.pl (I.J.-M.); ewojnar@wum.edu.pl (E.W.); dagmara.mirowska-guzel@wum.edu.pl (D.M.-G.); 5Department of Invasive Cardiology, Central Clinical Hospital of the Ministry of Interior and Administration, Wołoska 137, 02-507 Warsaw, Poland; 6Interdisciplinary Centre for Mathematical and Computational Modeling, University of Warsaw, Adolfa Pawińskiego 5A, 02-106 Warsaw, Poland; j.zielinski@icm.edu.pl

**Keywords:** sirtuin 1, IL-27, visfatin, female, type 1 diabetes mellitus, thyroid autoimmunity, cardiovascular disease

## Abstract

The loss of cardioprotection observed in premenopausal, diabetic women may result from the interplay between epigenetic, metabolic, and immunological factors. The aim of this study was to evaluate the concentration of sirtuin 1, visfatin, and IL-27 in relation to cardiovascular parameters and Hashimoto’s disease (HD) in young, asymptomatic women with type 1 diabetes mellitus (T1DM). Thyroid ultrasound, carotid intima-media thickness (cIMT) measurement, electrocardiography, and echocardiography were performed in 50 euthyroid females with T1DM (28 with HD and 22 without concomitant diseases) and 30 controls. The concentrations of serum sirtuin 1, visfatin and IL-27 were assessed using ELISA. The T1DM and HD group had higher cIMT (*p* = 0.018) and lower left ventricular global longitudinal strain (*p* = 0.025) compared to females with T1DM exclusively. In women with a double diagnosis, the sirtuin 1 and IL-27 concentrations were non-significantly higher than in other groups and significantly positively correlated with each other (*r* = 0.445, *p* = 0.018) and thyroid volume (*r* = 0.511, *p* = 0.005; *r* = 0.482, *p* = 0.009, respectively) and negatively correlated with relative wall thickness (*r* = –0.451, *p* = 0.016; *r* = –0.387, *p* = 0.041, respectively). These relationships were not observed in the control group nor for the visfatin concentration. These results suggest that sirtuin 1 and IL-27 contribute to the pathogenesis of early cardiac dysfunction in women with T1DM and HD.

## 1. Introduction

The accelerated atherosclerosis and early development of heart failure (HF) observed in diabetes patients have a multifactorial background and are still insufficiently understood [1]. In recent years, increasing attention is being paid to sex differences in cardiovascular (CV) risk, which in combination with diabetes leads to a loss of cardioprotection observed in premenopausal women. However, the mechanisms by which sex may modulate the effect of diabetes on CV risk have not been exhaustively investigated.

It has been shown in large meta-analyses that there is a significantly higher risk of coronary heart disease [2] and HF associated with diabetes in women than men, especially in type 1 diabetes mellitus (T1DM) [3]. In fact, T1DM was associated with a 47% greater risk of HF in females than males as opposed to a 9% greater risk in type 2 diabetes mellitus (T2DM) [3]. Additionally, a registry-based cohort study confirmed that the onset of T1DM before the age of 10 years results in a higher loss of life years by 3.5 years in women than men [4]. Indeed, in light of the above studies, the latest guidelines from the European Society of Cardiology (ESC) in collaboration with the European Association for the Study of Diabetes (EASD) indicate the need for further investigation explaining the excess CV risk in females, especially with early-onset T1DM [5].

According to current knowledge, the loss of cardioprotection in diabetic women may be due to genetic and epigenetic mechanisms [6], differences in anatomical structure and metabolism of adipose tissue [7], hormone-related oxidative stress [8], and the gut microbiome connected with inflammation [9] and molecular factors involved in the formation of AGEs [10]. Another possible explanation is that there is a link between CV diseases and autoimmunity, potentially through inflammatory pathways, especially in T1DM [11]. Furthermore, it appears that the coexistence of a second autoimmune condition, of which Hashimoto’s disease (HD) is the most common, may be associated with increased CV risk [12]. Nevertheless, the exact mechanism of this phenomenon remains unclear.

As shown in Figure 1, the suspected link between epigenetic, metabolic, and immunological pathways may be sirtuin 1, an NAD+−dependent deacetylase that belongs to the enzymes of longevity [13,14].

Sirtuin 1, the epigenetic enzyme that regulates various cellular pathways, has attracted significant attention over recent years because it is involved in the pathogenesis of both CV diseases [24] and autoimmune disorders [25]. A recent meta-analysis of preclinical studies demonstrated that resveratrol and metformin reduce hyperglycemia, dyslipidemia, and insulin resistance, and exhibits a pro-inflammatory response by increasing the sirtuin 1-dependent intracellular antioxidant capacity [26]. Thus, it appears that sirtuin 1, due to its antioxidant properties, may be important in alleviating diabetic complications. Moreover, the ability of sirtuin 1 to regulate interleukin 27 (IL-27) expression probably determines the maintenance of the immune balance [27] and may be involved in the development of T1DM [28]. IL-27 can exert both pro-inflammatory and anti-inflammatory effects, which is currently being considered in novel therapeutic approaches for cardiovascular [29] and autoimmune diseases [30]. Another insufficiently explored molecule related to sirtuin 1 is visfatin [31]. This adipokine exhibits nicotinamide phosphoribosyltransferase (NAMPT) activity, and as an intracellular form (iNAMPT) maintains the activity of NAD-dependent enzymes, such as sirtuin 1 [19], while the extracellular form (eNAMPT) can induce inflammation and endothelial dysfunction [32]. Recently, this adipokine was suggested to be a therapeutic target in cardiovascular-metabolic disorders [33]. Despite the known interaction between visfatin and sirtuin 1 in the regulation of many processes responsible for cell metabolism and control of circadian rhythm [34], these relationships are still not well understood, especially in relation to T1DM.

Given the often atypical or clinically silent progression of CV diseases in women with diabetes, identification of specific risk factors is a key step in early diagnosis and effective prevention. Thus, the purpose of this pilot study was to evaluate the potential utility of sirtuin 1, visfatin, and IL-27 as markers of early development of atherosclerosis or HF in young, asymptomatic women with T1DM and HD. To the best of our knowledge, no research to date has assessed these serum levels in this context. The most relevant finding of our study is the demonstration of a positive correlation between sirtuin 1 and IL-27 levels and their significant association with thyroid and echocardiographic parameters in women with T1DM and HD.

## 2. Materials and Methods

### 2.1. Study Population

Fifty patients with T1DM were enrolled at the Diabetes Outpatient Clinic of the Central Clinical Hospital of the Ministry of Interior and Administration in *Warsaw, Poland*. Inclusion criteria were a diagnosis of T1DM, HbA1c < 10%, age 18–37 years, female, and current euthyreosis status in laboratory tests. Patients were excluded if they were pregnant or breastfeeding, had a positive history of CV diseases (hypertension, coronary heart disease, arrhythmia, HF), were smokers, or had hepatic or renal disorders. Women taking any additional medication apart from insulin treatment and levothyroxine were also excluded. The participants were divided into two study groups: group 1 (*n* = 28) included women diagnosed with both T1DM and HD, while group 2 (*n* = 22) included women diagnosed with T1DM without other diseases. The diagnosis of Hashimoto’s thyroiditis was stated on the basis of the standard criteria: Elevated thyroid antibodies (aTPO and/or aTG) titer and typical ultrasound. Among the female patients with HD, 18 were treated chronically with levothyroxine, and 10 had elevated antibodies and a typical ultrasound image indicating thyroid autoimmunity but with normal TSH and fT4 without medication.

Thirty age-matched, healthy women recruited from hospital employees and their relatives formed the control group (*n* = 30). The inclusion criteria were as follows: normal physical examination and no history of diabetes, CV diseases, autoimmune disorders, or other diseases. All the patients in the control group were without obvious features of thyroid autoimmunity during the ultrasound and had euthyreosis status in laboratory tests. Women taking any kind of medication, or who were pregnant or breastfeeding, were excluded.

The study protocol was prepared in accordance with the Declaration of Helsinki and was approved by the local Ethical Committee (Approval of the Ethics and Surveillance Committee for Research in Human and Animal Sciences at the Central Clinical Hospital of the Ministry of Internal Affairs; No. 22/2018 of 09 May 2018). Each patient provided informed written consent to participate in this study.

### 2.2. Clinical Data Collection

Medical history and anthropometric measures were recorded during patient interviews. All the enrolled patients underwent a physical examination. The BMI (body mass index) was estimated as body weight divided by the squared height (kg/m^2^). Basic data were collected from each individual on a computerized datasheet.

### 2.3. Echocardiography and Electrocardiography

Complete 2-dimensional echocardiograms and Doppler scan studies were performed and analyzed by an expert echocardiographer following the recommendations of the European Association of Cardiovascular Imaging [35] using an EPIQ system (version 7C, Philips Medical Systems, Best, The Netherlands. The observer was blinded to the group under examination.

The left ventricular dimension (LVD), interventricular septum (IVS), and posterior wall dimensions (PWD) in systole and diastole were measured using M-mode by the parasternal long-axis view. Atrial volumes and left ventricular ejection fraction (EF) were assessed using Simpson’s method [35]. The left atrial volume index (LAVI) was calculated by dividing the left atrium (LA) volume (measured from standard apical two- and four-chamber views at end-systole) by the body surface area of participants. The left ventricular mass (LVM) was calculated using the Devereux formula indexed to body surface area (left ventricular mass index (LVMI)) [36]. The relative wall thickness (RWT) was calculated as double the PWD divided by the left ventricle diastolic diameter [37]. E/A ratios were calculated using pulse-wave Doppler early (E) and late (A) diastolic mitral peak velocities together with the isovolumic relaxation time (IVRT) and deceleration time (DT). Pulse-wave Tissue Doppler Imaging was used to measure peak diastolic velocities of the septal (E’spt, A’spt) and lateral (E’lat, A’lat) corner of mitral annulus in apical four-chamber view. Two-dimensional speckle tracking analyses was performed in the longitudinal three-chamber, two-chamber, and four-chamber views for left ventricular global longitudinal strain (GLS) [35]. A 12-lead standard electrocardiography ECG (10 mm = 1 mV, 25 mm/s) was performed in a supine position during rest. The QT interval was corrected (QTc) for heart rate using the Bazett formula [38].

### 2.4. Measurement of Carotid Intima-Media Thickness and Thyroid Assessment

Carotid arteries were examined bilaterally using B-mode ultrasonography with a 10 MHz linear transducer and this procedure was performed by the same sonographer according to the guidelines of the Polish Ultrasonography Society [39]. CIMT was measured three times on each side (right and left) and the mean value was calculated for each side and for all six measurements included in the analyses. All participants underwent thyroid ultrasonography according to the guidelines of the Polish Ultrasonography Society [40]. The thyroid volume was calculated using a simplified formula for the volume of a spheroid: V = 0.5 × W × H × L, where V is the volume of the lobe, 0.5 is the simplified coefficient, W is the width, H is the height, and L is the length. The volume of the thyroid gland considered in the study was the sum of the volumes of the right and left lobes.

### 2.5. Biochemical Analysis

For laboratory tests, venous blood was collected after overnight fasting and included measurements of glycated hemoglobin (HbA1c), total cholesterol (TC), high-density lipoprotein (HDL), low-density lipoprotein (LDL), triglyceride (TG) levels, and thyroid parameters. The range of normal values for fT4 was between 0.93 and 1.7 ng/dL, for fT3 it was between 2.0 and 4.4 pg/mL, and for TSH it was between 0.27 and 4.2 µIU/mL. The positive values for antibodies were >34 IU/mL for anti-TPO-Abs and >115 IU/mL for anti-TG-Abs. All of the collected blood samples were subjected to biochemical analyses immediately after collection on an ongoing basis in the hospital laboratory using standard methods. For all participants, sera were separated by centrifugation of the whole blood samples for 10 min and were stored at −80 °C for further measurement of immune markers.

### 2.6. Sirtuin 1, Visfatin, and IL-27 Measurements

Serum concentrations of sirtuin 1, visfatin, and IL-27 were analyzed using commercial immunoenzymatic assays by an experienced laboratory technician in the Department of Experimental and Clinical Pharmacology (Medical University of Warsaw, Warsaw, Poland). The IL-27 serum concentration was assessed in duplicate by the enzyme-linked immunosorbent assay (ELISA) kit (Human IL-27 ELISA Kit, Covalab, Bron, France; sensitivity: 12.8 pg/mL; reference range: 0–1000 pg/mL). The optical density was assessed at 450 nm using a BIOTEK spectrophotometric reader (EPOCH model, Winooski, VT, USA.). The visfatin serum level was determined in duplicate using a commercial ELISA kit (Human PBEF/Visfatin ELISA Kit, Abclonal, Wuhan, China; sensitivity: 0.55 ng/mL; reference range: 0–100 ng/mL). Optical density was assessed at 450 nm using spectrophotometric reader BIOTEK (model EL×800, Winooski, VT, USA). Finally, the serum concentration of sirtuin 1 was measured by competitive ELISA (Human SIRT1 (Sirtuin 1) ELISA Kit, FineTest, Wuhan, China; sensitivity: 0.188 ng/mL; reference range: 0.313–20 ng/mL). Optical density was assessed at 450 nm using a BIOTEK spectrophotometric reader (model EL×800, Winooski, VT, USA). Measurements for sirtuin 1 serum concentration were performed in a single replicate based on dilution tests. All procedures were performed in accordance with the manufacturer’s recommendations.

### 2.7. Statistical Analysis

Data were analyzed using Statistica 13. The Lilliefors and Shapiro–Wilk tests were used to verify the normal distribution of data. A non-parametric Mann–Whitney U test was used to compare two groups of unrelated variables. Differences between multiple groups were evaluated using a Kruskal–Wallis ANOVA test. To explore correlations among variables, a Spearman’s correlation (r) test was used. Statistical significance was set at a *p*-value < 0.05.

## 3. Results

### 3.1. Study Population

The study population consisted of 80 females (mean age 26.4 ± 4.4 years and range 18–37 years) including 30 controls and 50 participants with T1DM divided into two groups: T1DM and HD (T1DM + HD, *n* = 28) and T1DM without any other additional diseases (T1DM – HD, *n* = 22) (Table 1).

There were no significant differences found in age, sex, body mass index (BMI), lipid parameters, TSH and fT4 levels, heart rate, or QTc interval among the sample groups. Importantly, women in all study groups had normal BMI and a favorable lipid profile. Notable, the rather low levels of LDL and TG, and fairly high levels of HDL present in the group with T1DM would generate a low atherogenicity index. Nevertheless, the group with T1DM had significantly thicker cIMT (*p* = 0.046) and lower EF (*p* = 0.001) compared to the control group. One explanation for these differences may be the association between historic glycemic exposure and its ‘metabolic memory’ and the long-term CV risk [41]. It appears that non-enzymatic glycation of proteins and lipids leads to the activation of oxidative stress, causing long-term CV complications despite the attainment of glycemic control, which has been reported in T1DM [42]. Indeed, the metabolic memory phenomenon results from the dysregulation of epigenetic mechanisms that lead to changes in the expression of pathological genes responsible for chronic inflammation, which plays a key role in the development of atherosclerosis or cardiac dysfunction [43].

The groups of patients with T1DM were similar in diabetes duration, HbA1c, daily insulin requirement, lipid parameters, and thyroid hormones. However, women with T1DM and HD exhibited significantly thicker cIMT (*p* = 0.018) and lower left ventricular GLS (*p* = 0.025) compared to women with T1DM only (Figure 2). These observed differences may be due to the relationship between autoimmunity and CV risk, as further explained in the discussion. Among the echocardiographic parameters considered, we found significantly lower LVMI (*p* = 0.01) and higher IVRT (*p* < 0.0001) in women with T1DM compared with the control group, and lower LAVI (*p* = 0.032) and higher RWT (*p* = 0.04) in women with T1DM and HD compared to women with T1DM only.

### 3.2. Circulating Sirtuin 1, IL-27 and Visfatin Levels

Although there were no significant differences in the concentration of the markers tested between the sample groups, we highlight the differences that did exist. The levels of sirtuin 1 were non-significantly higher in patients with T1DM compared to controls (19.51 ± 65.70 ng/mL vs. 17.51 ± 38.39 ng/mL). Dividing T1DM patients into two groups according to the presence (T1DM + HD) or absence of HD (T1DM − HD), a higher sirtuin 1 concentration was found in the T1DM + HD group (32.68 ± 86.06 ng/mL vs. 2.76 ± 4.88 ng/mL. The sirtuin 1 concentration was lower in patients with T1DM alone compared with the control group. These results suggest a possible association of elevated sirtuin 1 concentration with patients that have T1DM and HD (Figure 3A).

The levels of IL-27 were also non-significantly higher in women with T1DM compared to controls (22.06 pg/mL vs. 14.5 pg/mL). Similar to sirtuin 1, the levels of IL-27 were highest in the group with T1DM and HD (27.82 pg/mL) (Figure 3B). We found a significant positive correlation between sirtuin 1 and IL-27 levels in patients with T1DM (*R* = 0.329, *p* = 0.02) and with T1DM and HD (*R* = 0.445, *p* = 0.018). There was no significant association between sirtuin 1 and IL-27 levels in controls. In turn, the levels of visfatin were non-significantly lower in women with T1DM compared to controls (14.50 ± 47.82 ng/mL vs. 22.06 ± 60.38 ng/mL). In the T1DM + HD group, the visfatin concentration was even lower than in women with T1DM only (8.87 ± 37.47 ng/mL vs. 21.66 ± 58.61 ng/mL) (Figure 3C). We found no correlation in the serum levels of visfatin and sirtuin 1 or visfatin and IL-27 in any of the groups.

### 3.3. Correlation Analysis between the Circulating Markers and Other Studied Variables Using Spearman’s Correlation (r) (Supplementary Tables S1–S3)

We explored the correlation between sirtuin 1 (Appendix A), IL-27 (Appendix A), and visfatin (Appendix A) levels with known CV risk factors (duration of diabetes, HbA1c, BMI, lipid parameters), thyroid parameters (TSH, fT4, fT3, anti-TPO, anti-TG, thyroid volume), and selected CV parameters (cIMT thickness, QTc interval, resting HR, echocardiographic parameters). All analyses were performed in four groups: group 1 + 2 included all women with T1DM (*n* = 50), group 1 comprised women with T1DM and HD (*n* = 28), group 2 comprised women with T1DM and no other conditions (*n* = 22), and group 3 contained healthy women as controls (*n* = 30).

Spearman’s correlation tests revealed that there were no significant correlations between sirtuin 1, visfatin, and IL-27 serum levels with duration of diabetes, total daily dose of insulin, and HbA1c levels. The visfatin level was positively associated with BMI, but only in the control group (*R* = 0.370, *p* = 0.044), and was negatively correlated with total cholesterol in group 1 (T1DM + HD) (*R* = −0.375, *p* = 0.049). With regard to the lipid parameters, there was a negative correlation between triglyceride levels and IL-27 (*R* = −0.437, *p* = 0.042) and sirtuin 1 levels (*R* = −0.489, *p* = 0.021) in women with T1DM only (group 2). IL-27 levels were negatively associated with LDL levels in women with T1DM only (*R* = −0.424, *p* = 0.05). These findings suggest a possible role of sirtuin 1, visfatin, and IL-27 in the lipid balance of women with T1DM.

Considering thyroid parameters, women with T1DM and HD (group 1) exhibited a strong positive correlation between thyroid volume and both sirtuin 1 (*R* = 0.511, *p* = 0.005) and IL-27 levels (*R* = 0.482, *p* = 0.009). Similar, although less pronounced associations were found in all women with T1DM, in which IL-27 was positively correlated with aTG (*r* = 0.331, *p* = 0.019). In turn, visfatin levels were positively associated with aTPO in all women with T1DM (*r* = 0.321, *p* = 0.023) and were negatively correlated with fT4 levels in group 2 (T1DM – HD) (*R* = –0.591, *p* = 0.004). These results suggest that higher visfatin concentrations in the group of women with T1DM but without HD may be associated with reduced thyroid endocrine function and perhaps a greater predisposition to the development of autoimmune thyroid disease.

The electrocardiographic parameters QTc interval and HR at rest were negatively correlated with sirtuin 1 in the T1DM – HD group (*R* = –0.435, *p* = 0.043 and *R* = –0.506, *p* = 0.016, respectively); in the control group, the QTc interval was significantly positively associated with sirtuin 1 (*R* = 0.510, *p* = 0.004). These differences may be due to the varied effects of sirtuin on the CV system depending on metabolic disturbances. Our results showed no significant correlation between either visfatin or IL-27 levels and QTc interval.

cIMT was positively correlated with visfatin levels in the control group (*R* = 0.447, *p* = 0.013). There was no correlation between cIMT and study markers in women with T1DM. Our results suggest a possible association of visfatin with subclinical carotid artery atherosclerotic lesions, but this relationship does not appear to be specific to T1DM.

Among the echocardiographic parameters studied, we found a significant negative correlation between RWT and PWD with sirtuin 1 (*R* = −0.451, *p* = 0.016; and *R* = −0.522, *p* = 0.004, respectively) in women with T1DM and HD (group 1). In patients with T1DM only (group 2), sirtuin 1 was correlated with wave E, E/A’med, and A’med (*R* = −0.440, *p* = 0.040; *R* = 0.426, *p* = 0.048; *R* = −0.478, *p* = 0.024, respectively). In the control group there was one significant correlation between sirtuin 1 and A’med (*R* = 0.391, *p* = 0.033). Thus, sirtuin 1 levels appear to be correlated with echocardiographic parameters related to diastolic function, specifically in women with T1DM or a double diagnosis.

IL-27 was correlated with RWT in patients with T1DM and HD (*R* = −0.389, *p* = 0.041), but this correlation was not observed in the other groups, which suggests an association of this cytokine with changes associated with myocardial hypertrophy in women with a double diagnosis. In contrast, visfatin levels were positively correlated with DT time (*R* = 0.424, *p* = 0019) in the control group, suggesting an association between this adipokine and early myocardial dysfunction, irrespective of the presence of T1DM.

## 4. Discussion

In this study, we focused on young asymptomatic women with T1DM, who were divided into groups according to the co-occurrence of HD. As recent reports indicate, this diabetic group is at high risk of developing CV complications [2,3,4]. Although it is widely recognized that patients with autoimmune diseases are in a higher CV risk group, the molecular factors involved in this process are not well-known. A possible hypothesis is that this involves an interaction between epigenetic, metabolic, and immunological factors.

Our results show that women with T1DM and HD, despite laboratory euthyroidism, have more unfavorable CV parameters (i.e., significantly thicker cIMT and lower left ventricular GLS) compared to women with T1DM without HD. Such findings suggest the involvement of an autoimmune component connected with autoimmune thyroid disease in the development of CV complications. In order to gain a deeper understanding of the pathogenesis of CV disease in women, we also assessed sirtuin 1, IL-27, and visfatin levels in the study population and we checked their correlation with selected CV parameters. Despite finding no significant differences in marker concentrations between groups, we found that women with a double autoimmune diagnosis had higher sirtuin 1 and IL-27 levels and lower visfatin serum levels than women with T1DM only, or women in the control group. Moreover, the sirtuin 1 was positively correlated with IL-27 levels in women with T1DM and HD, which suggests an association with thyroid autoimmunity. Indeed, it has been shown that sirtuin 1 interacts with transcription factor IRF-1 (Interferon Regulatory Factor 1) at the molecular level, leading to the regulation of IL-27 expression, which is important in Th17 differentiation [44]. In turn, Th17 lymphocytes have been shown to play an essential role in the induction of organ-specific autoimmune diseases including chronic lymphocytic thyroiditis [45]. Our data suggest a possible detrimental effect of IL-27, which is likely to contribute to the development of autoimmune thyroid disease in individuals with T1DM. In contrast, a recent study found reduced IL-27 serum levels in patients with Graves’ disease, suggesting a possible anti-inflammatory role of these cytokines [46]. A possible explanation for these differences is the effect of IL-27 on the Th1 to Th2 ratio [47], which is dominant in HD, whereas it is non-dominant in Graves’ disease [48]. IL-27 most likely enhances Th1 lymphocyte function and suppresses Th2, which would suggest an immune imbalance similar to that seen in T1DM or HD [49]. In line with this assumption, studies of single nucleotide polymorphisms have confirmed an association between IL-27 [50] and the SIRT1 gene with autoimmune thyroid diseases [51]. In this study, aTG was positively correlated with the IL-27 serum levels in women with T1DM, but not significantly with sirtuin 1 serum levels. However, both markers were significantly associated with thyroid volume, which may be connected to the autoimmune connective tissue proliferation. Therefore, an evaluation of the local concentration of sirtuin 1 and IL-27 in thyroid tissue should be considered as a future research direction.

Based on our findings, the marker most associated with echocardiographic parameters related to early cardiac dysfunction in women with T1DM is sirtuin 1. Interestingly, a significant negative correlation between sirtuin 1 with RWT and PWD was only found in women with T1DM and HD, which suggests an association between this marker with the left ventricular remodeling in this group of patients. Our results indicated that higher sirtuin 1 levels were correlated with reduced RWT, which on the one hand may be beneficial, but on the other can be associated with increased LVMI, which may suggest a higher risk of eccentric myocardial hypertrophy. As was suggested by previous studies, patients with T1DM are more likely than those with T2DM to have the dilated type of cardiomyopathy with systolic cardiac dysfunction, which is characterized by eccentric myocardial hypertrophy [52]. Notably, such cardiac remodeling is probably related to autoimmunity [53]. Moreover, the observed differences in the phenotype of HF depending on the type of diabetes have been explained by experimental studies in animal models, indicating different autophagy activity in cardiomyocytes, which was increased in subjects with T1DM and decreased in those with T2DM [54]. Indeed, the greater predisposition to developing HF with a dilated phenotype observed in longstanding T1DM is associated with cardiomyocyte loss and progressive fibrosis, which may have an autoimmune basis [55]. However, data on the assessment of sirtuin 1 and IL-27 in relation to CV parameters in autoimmune diseases are limited. The results of studies conducted in patients with T2DM were contradictory [56,57,58]. Nonetheless, a growing body of evidence suggests that increasing the sirtuin 1 expression using antidiabetic drugs, such as metformin [59] or SGLT2 inhibitors [60], may have a beneficial effect on the CV system. At the molecular level, it has been shown that this is associated with a strong antioxidant and anti-inflammatory effect by interacting with signaling pathways related to nuclear factor-kappa β (NF-κB), mitochondrial adapter p66Shc, and forehead box O (FoxO) [24]. Sirtuin 1 probably reverses vascular endothelial dysfunction by inhibiting NF-κB transcriptional activity, which mediates the production of inflammatory cytokines and prothrombotic markers [61]. Blood vessels can be protected from hyperglycemia-induced endothelial dysfunction by reducing the expression of p66Shc, which directly stimulates mitochondrial generation of reactive oxygen species [62]. Moreover, accumulating evidence supports the proposal that sirtuin 1 increases endothelial nitric oxide synthase and upregulates FoxO, activating its antioxidant properties and delaying cellular aging. Furthermore, sirtuin 1 is believed to inhibit the release of proinflammatory adipokines from dysfunctional perivascular adipose tissue that mediates vascular calcification [63]. Sirtuin 1 also exerts cardioprotective effects by reducing endoplasmic reticulum stress against myocardial infarction injury [64,65]. In conclusion, the protective properties of sirtuin 1 against oxidative stress reduce endothelial dysfunction, inflammation, remodeling of arterial walls, vascular aging, atherosclerosis, and heart damage [24]. It has been suggested by experimental studies that the cardioprotective effect of sirtuin 1 depends on its concentration: A mild to moderate expression of sirtuin 1 exerted a beneficial effect, inhibiting aging of the heart, whereas a high level of sirtuin 1 increased oxidative stress and cardiac damage [66]. Furthermore, one recently published study has shown that sirtuin 1 activity in circulating peripheral blood mononuclear cells (PBMCs) can be a biomarker of different HF phenotypes, with increased levels associated with a loss of myocardial systolic function [67]. Although in this study we measured serum sirtuin 1 levels in asymptomatic patients, our results gave similar observations. Indeed, higher serum sirtuin 1 levels were present in T1DM and HD patients who had lower left ventricular GLS, which is an early marker of cardiac systolic dysfunction. We found a significant positive correlation between IL-27 and RWT in women with T1DM and HD. This observation is in accordance with previous studies that showed an association between the IL-27 gene polymorphism involving SNP rs153109 with a predisposition to dilated cardiomyopathy [68] and the deterioration of left ventricular function in the course of acute coronary syndrome [69]. Our findings confirm the need for further research of the role of IL-27 in the pathogenesis of HF and have for the first time highlighted its association with sirtuin 1 levels.

We found a significant negative correlation between sirtuin 1 levels and the QTc interval in women with T1DM and a positive correlation for the same parameters in women in the control group. Hence, it appears that sirtuin 1 may exert both beneficial and negative effects on the development of CV complications depending on its concentration, local expression, and individual variability. Kedenko et al. demonstrated associations between genetic variations at the SIRT1 with carotid atherosclerosis, with about 3–4 times higher effect sizes for women than men [70]. However, we found no significant association between sirtuin 1 serum levels and cIMT in all groups of women. Similar results were obtained for IL-27, although its pathogenic role in the development of atherosclerotic disease has been indicated [71]. We did find an association between lipid parameters and IL-27 and sirtuin 1 levels in women with T1DM, which may lead to an increased risk of atherosclerosis. It was demonstrated in an experimental study that atherogenic dyslipidemia enhances autoimmune responses in a Toll-like receptor 4 and IL-27-dependent manner [23]. By contrast, in our study, women with T1DM had a favorable lipid profile, suggesting a low atherogenic index. Despite this, we found significantly higher cIMT and lower EF in the group of women with T1DM compared to the control group, and a significantly higher cIMT and lower left ventricular GLS in the group of women with T1DM and HD compared to women with T1DM exclusively. The hypothesis we propose to explain our results is that there is an association between autoimmunity and CV risk. Furthermore, increasing evidence suggests that inflammatory immune responses in patients with T1DM or HD are not restricted to the pancreas or thyroid gland, instead of causing systemic inflammation. [72,73]. It is important to note that our study was conducted with women only, while the immune response that increases inflammation through various signaling pathways is likely to depend on sex differences [74]. Sex steroids have been shown to be able to switch immune cells from anti-inflammatory to pro-inflammatory actions and to regulate mitochondrial production of reactive oxygen species. While in healthy women estrogens have the ability to reduce oxidative stress, in the presence of autoimmune disease this benefit diminishes, which could explain the increased CV risk in T1DM women [75]. However, the exact mechanism governing these relationships is not known, and there is a need for further research into the molecular basis of CV disease progression in autoimmune disorders.

Given the association of sirtuin 1 with visfatin at the cellular level and the recognition of this adipokine as a key mediator in the interaction between epigenetic and metabolic factors in CV complications, its serum levels were assessed in our study. Based on previous studies, visfatin has been identified as a potential biomarker of insulin resistance, obesity, metabolic syndrome, type 2 diabetes mellitus, and CV disease [76]. Moreover, recent reports have indicated its association with the development of inflammation in autoimmune inflammatory disorders [20,77,78,79]. However, results concerning visfatin in T1DM and/or HD are limited and conflicting. In previous studies, circulating visfatin levels in T1DM were higher [80,81] or lower [82,83,84] compared to healthy controls. Recent studies have suggested a relationship between increased visfatin levels and chronic autoimmune thyroiditis [78,85]. Our study showed non-significantly lower serum levels of visfatin in women with T1DM compared with women in the control group. The coexistence of two autoimmune diseases (T1DM and HD) was associated with even lower visfatin levels compared with women diagnosed exclusively with T1DM or healthy participants, but these differences were also non-significant. Our results correspond with those obtained in a recent study by Tompa et al., in which children with a combined diagnosis of two autoimmune diseases (T1DM and celiac disease) had lower visfatin levels compared with children diagnosed exclusively with T1DM [86]. A possible mechanism underlying the negative correlation of visfatin with autoimmune dual diagnosis may be due to the function of visfatin in initiating de novo NAD biosynthesis, a deficiency of which is associated with a decreased amount of Treg [87]. It is suspected that the altered function of natural Treg cells results from epigenetic modification and may be connected with susceptibility to common autoimmune diseases [88] including autoimmune thyroid diseases [89]. However, the significant negative correlation found in our study between visfatin and ft4 in women with T1DM only, and the positive correlation with aTPO in all women with T1DM, suggests an adverse effect of visfatin on the development of autoimmune thyroid disease, which needs to be verified through further study.

With regard to the CV parameters, no significant correlation with visfatin serum levels was found in women with T1DM or with a diagnosis of T1DM and HD. The significant association between visfatin serum levels and cIMT in controls suggest that this adipokine may be involved in the progression of atherosclerosis, but without the link to autoimmunity. Visfatin probably affects the endothelial function and secretion of pro-inflammatory cytokines through the NF-κB pathway in the early stages of atherosclerotic plaque formation [79]. Notably, we also found a significant positive correlation between the deceleration time in controls, but not in T1DM women. This finding is consistent with previous studies, suggesting that an increased visfatin plasma level may be involved in the development of HF [90] or adverse CV events [91]. In fact, this adipokine, which is secreted by adipocytes and pro-inflammatory cells [92], probably exerts its effect on CV organs both through paracrine secretion in perivascular [93] and pericardial adipose tissue [94], as well as through its release into the circulation. Therefore, visfatin should be considered as a biomarker for the development of early atherosclerosis.

The main strength of our study is the inclusion of a homogeneous population of women, which allowed us to determine differences in CV parameters independent of poor metabolic control, hyperlipidemia, hypertension, or thyroid hormonal imbalance. In addition, this is the first study to determine serum sirtuin 1, IL-27, and visfatin levels in women with T1DM and HD in relation to CV parameters. Notably, parts of the study were conducted by independent, blinded, experienced investigators from different centers, which allowed a more objective summary of the results. However, this study also has some limitations: (1) a small sample of patients was included in each group, which may affect the results of the study; (2) a lack of local expression assessment of the markers in thyroid tissue or myocardium means we could not provide a complete insight of the interaction between the molecules; (3) we did not assess CV parameters and studied markers in women with HD only. Moreover, to establish the female specificity of the markers it would be necessary to compare their concentrations in male and symptomatic patients with T1DM and CV diseases. Data regarding diet and physical activity, which may alter the hormonal profile and outcomes associated with CVD and T1DM, could have improved the study. Nevertheless, the limitations of our pilot study indicate new directions for further research, providing a better understanding of the molecular mechanisms of CV complications specific to women with T1DM, which, in light of the high CV risk of this group of patients, is essential to discovering a new therapeutic approach.

## 5. Conclusions

In this study, young asymptomatic women with T1DM and HD, despite laboratory euthyroidism, had significantly higher cIMT thickness and lower left ventricular GLS compared to women with T1DM only or without T1DM (control group). Although we found no significant differences in serum concentrations of sirtuin 1, IL-27, or visfatin between groups, our results indicate that cardiac dysfunction in T1DM and HD females may be associated with higher levels of sirtuin 1 and IL-27, which were positively correlated with each other and with echocardiographic parameters. Based on these findings, we propose that an interaction between sirtuin 1 and IL-27 may be considered a biomarker of early cardiac dysfunction in women with T1DM and HD. However, further research with a larger sample size will be needed to assess the potential relevance of these serum levels and their effect on CV complications in T1DM patients.

## Figures and Tables

**Figure 1 biomolecules-11-01110-f001:**
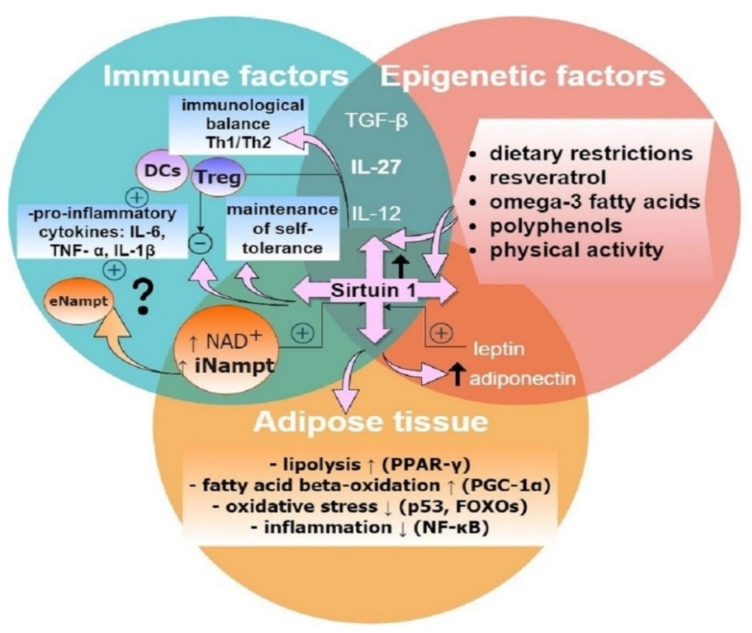
Sirtuin 1 is a possible link between metabolic and immunological pathways. The expression of sirtuin 1 depends on epigenetic factors and influences adipose tissue metabolism through various transcription factors to reduce lipogenesis [15] and oxidative stress [16], and it also has anti-inflammatory effects [17] and regulates the secretion of adipokines [18]. Visfatin intracellularly (iNampt) catalyzes the reaction of the NAD+ synthesis pathway, which increases sirtuin 1 activity [19], while extracellularly (eNampt) it regulates responses to oxidative stress and inflammation [20]. Sirtuin 1 influences the immune system through cytokines, maintaining the balance between Th1/Th2/Th17/Treg lymphocytes’ defense against autoimmunity [21]. IL-27, together with Treg lymphocytes, can exert an anti-inflammatory effect [22], protecting against autoimmune inflammation or a pro-inflammatory effect when secreted by dendritic cells (DCs), depending on the Toll-like receptor 4 (TLR 4) [23]. Abbreviations: DCs—dendritic cells, eNampt—extracellular nicotinamide phosphoribosyltransferase, FOXO—forehead box O, iNampt—intracellular nicotinamide phosphoribosyltransferase, NF-κB—nuclear factor κ-light-chain-enhancer of activated B cells, PGC1a—peroxisome proliferator-activated receptor c coactivator 1a, PPAR-γ—peroxisome proliferator-activated receptor gamma, TNF-α—tumor necrosis factor α, Treg—T regulatory lymphocytes.

**Figure 2 biomolecules-11-01110-f002:**
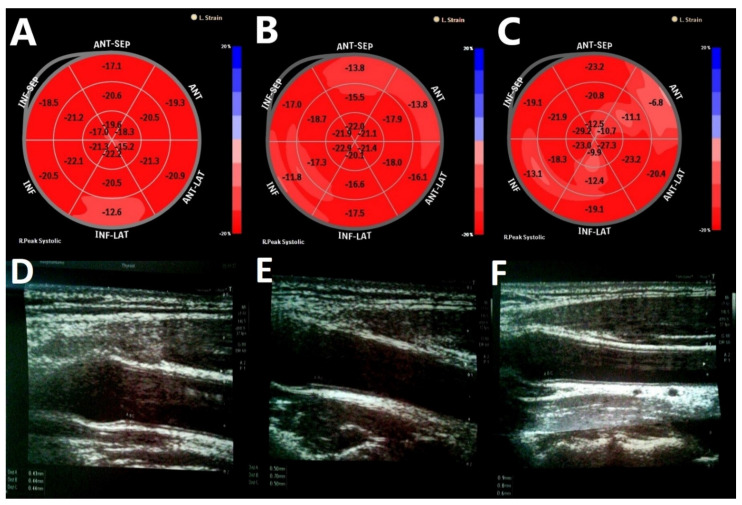
Left ventricular global longitudinal strain (GLS) in (**A**) a healthy subject, (**B**) women with exclusively type 1 diabetes mellitus (T1DM), (**C**) women with T1DM and Hashimoto’s disease (HD); and ultrasound images of carotid intima-media thickness (cIMT) measurement in (**D**) a healthy subject, (**E**) women with exclusively T1DM, and (**F**) women with T1DM and HD.

**Figure 3 biomolecules-11-01110-f003:**
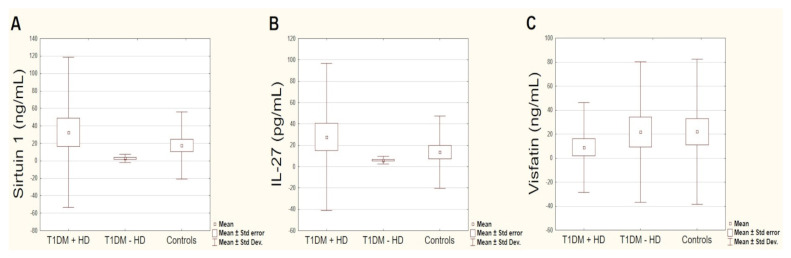
(**A**) Sirtuin 1 serum levels, (**B**) IL-27 serum levels, and (**C**) visfatin serum levels for each sample group in the study population.

**Table 1 biomolecules-11-01110-t001:** The main demographic and clinical characteristics of each test group.

	T1DM (*n* = 50)	T1DM + HD (*n* = 28)	T1DM − HD (*n* = 22)	Controls (*n* = 30)	*p* (T1DM vs. Controls)	*p* (T1DM + HD vs. T1DM − HD)
Age (years)	26.2 ± 4.75	26.68 ± 4.64	25.59 ± 4.92	26.7 ± 3.72	0.548	0.363
BMI (kg/m^2^)	22.45 ± 3.14	22.78 ± 3.29	22.04 ± 2.95	22.53 ± 3.03	0.941	0.481
Diabetes duration (years)	13.04 ± 6.44	14.43 ± 7.20	11.27 ± 4.93	0.00	0.000	0.143
Daily insulin dose (units)	38.48 ± 18.86	39.85 ± 21.50	36.73 ± 15.15	0.00	0.000	0.837
HD duration (years)	4.08 ± 5.99	7.29 ± 6.40	0.00	0.00	0.000	0.000
HbA1c (%)	7.7 ± 1.33	8.03 ± 1.35	7.28 ± 1.20	5.18 ± 0.28	0.000	0.071
Total cholesterol (mg/dL)	173.06 ± 28.27	174.57 ± 27.97	171.14 ± 29.19	172.77 ± 29.23	0.917	0.647
LDL (mg/dL)	86.36 ± 25.91	87.75 ± 28.61	84.59 ± 22.52	83.93 ± 28.36	0.735	0.822
HDL(mg/dL)	70.66 ± 19.35	70.75 ± 18.73	70.55 ± 20.54	72.57 ± 21.10	0.713	0.718
TG (mg/dL)	81.14 ± 33.32	80 ± 34.96	82.59 ± 31.86	82.73 ± 30.36	0.651	0.777
TSH (µIU/mL)	2.26 ± 1.95	2.53 ± 2.45	1.90 ± 0.92	2.44 ± 1.15	0.127	0.718
ft4 (ng/dL)	1.25 ± 0.23	1.28 ± 0.26	1.22 ± 0.18	1.23 ± 0.17	0.608	0.200
ft3 (pg/mL)	2.97 ± 0.46	2.94 ± 0.42	3.01 ± 0.52	3.26 ± 0.53	0.007	0.992
aTPO IU/mL	102.51 ± 175.89	174.70 ± 209.32	10.64 ± 7.68	24.85 ± 65.16	0.000	0.000
aTG IU/mL	144.51 ± 167.58	225.89 ± 168.98	40.95 ± 93.53	80.88 ± 132.32	0.004	0.000
Thyroid volume (mL)	11.48 ± 3.55	11.38 ± 3.94	11.61 ± 3.08	11.52 ± 3.82	0.626	0.740
EKG-HR	72.86 ± 11.35	72.5 ± 10.56	73.32 ± 12.53	71.53 ± 10.16	0.673	0.992
EKG-Qtc	426.38 ± 19.51	427.71 ± 19.45	424.68 ± 19.91	421.27 ± 19.76	0.220	0.853
cIMT (mm)	0.66 ± 0.28	0.72 ± 0.34	0.58 ± 0.12	0.58 ± 0.10	0.046	0.018
EF (%)	62.27 ± 3.63	62.34 ± 3.40	62.19 ± 3.99	64.53 ± 2.03	0.001	0.399
LAVI mL/m^2^	22.95 ± 5.3	21.51 ± 5.19	24.78 ± 4.97	24.29 ± 5.88	0.398	0.032
LVEDd (mm)	42.9 ± 4.16	42.11 ± 4.55	43.91 ± 3.45	44.57 ± 2.45	0.089	0.151
IVSD (mm)	8.56 ± 1.46	8.64 ± 1.68	8.45 ± 1.14	9 ± 1.31	0.114	0.992
PWD (mm)	8.92 ± 1.14	9.21 ± 1.20	8.55 ± 0.96	8.93 ± 0.98	0.858	0.082
LVMI g/m^2^	69.02 ± 11.73	68.79 ± 11.76	69.32 ± 11.97	76.68 ± 10.16	0.010	0.792
RWT	0.42 ± 0.10	0.45 ± 0.12	0.39 ± 0.04	0.40 ± 0.04	0.592	0.040
A wave (ms)	58.01 ± 14.41	59.61 ± 12.16	55.98 ± 16.92	60.45 ± 15.72	0.499	0.324
E’sep (cm/s)	12.34 ± 2.21	12.11 ± 2.24	12.63 ± 2.17	13.20 ± 3.16	0.127	0.122
DT (ms)	215.72 ± 42.66	211.86 ± 41.78	220.63 ± 44.23	210.47 ± 40.70	0.945	0.384
IVRT (ms)	94.5 ± 13.14	96.61 ± 13.66	91.82 ± 12.21	79.87 ± 8.98	0.000	0.278
LV GLS	17.59 ± 1.97	16.90 ± 2.12	18.32 ± 1.53	18.96 ± 2.50	0.054	0.025
Sirtuin 1 (ng/mL)	19.51 ± 65.70	32.68 ± 86.06	2.76 ± 4.88	17.51 ± 38.39	0.106	0.278
Visfatin (ng/mL)	14.50 ± 47.82	8.87 ± 37.47	21.66 ± 58.61	22.06 ± 60.38	0.347	0.097
IL-27 (pg/mL)	18.23 ± 52.33	27.82 ± 68.87	6.05 ± 3.56	13.47 ± 33.88	0.622	0.132

## Data Availability

The data presented in this study are available on request from the first author. The data are not publicly available due to privacy restrictions.

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
