# Peer review of "Sirtuin 1, Visfatin and IL-27 Serum Levels of Type 1 Diabetic Females in Relation to Cardiovascular Parameters and Autoimmune Thyroid Disease"

_biomolecules, 2021, doi:10.3390/biom11081110_

Round 1
Reviewer 1 Report
In this study the authors investigated whether serum levels of Sirtuin1, IL-27 and visfatin were associated with cardiovascular parameters in T1D patients with or without Hshimoto's disease (HD).
Comments
1) To investigate the effect of HD, HD patients without T1D should be included.
2) In general, Sirtuin1 has anti-oxidant properties. From this standpoint, observations in this study should be discussed thoroughly.
3) Is total daily dose of insulin associated with Sirtuin1, IL-27 and visfatin levels?
Reviewer 2 Report
The manuscript submitted for publication by Łukawska-Tatarczuk et al., titled: “Sirtuin 1, Visfatin and IL-27 Serum Levels of Type 1 Diabetic Female in Relation to Cardiovascular Parameters and Autoimmune Thyroid Disease” is aiming to assess the relationship between Sirtuin 1, visfatin and IL-27 serum levels in T1DM and CVD parameters as well as autoimmunity particularly HD.
The manuscript is well written and presented and makes good contribution to the literature. A few points for consideration from the reviewer are listed below:
- Please use SI system consistently with commas ”,” versus points “.” When identifying increments examples 1,23 should be 1.27, etc.
- Table 1 column 1 T1DM (n=50) please correct the sign and introduce “=”.
- Table 1 results: The levels of LDL are rather low and HDL fairly high. This would generate a low atherogenicity index thus low risk for CVD counter to the clinical data reported. Along with the low lipemia these results are counterintuitive as per risk for CVD. Authors are invited to discuss these findings in greater detail and propose theories supporting their findings.
- Provide characteristic images of the clinical work performed indicating the differences in plaque thickness etc.
- The authors do not seem to consider nutritional/dietary aspects as well as exercise. Those can both alter hormonal profile and outcomes as related to CVD and T1DM. Did the authors perform nutritional assessment of their participants? Did they perform physical activity assessment? If not what was the rationale for excluding those aspects?
- Please use the term “participant” as opposed to “subject”.
Round 2
Reviewer 1 Report
I have no further comments.
Reviewer 2 Report
The reviewer believes that the authors have adequately and reasonably addresses his points on the manuscript and thus recommends publication.